# Metallacarborane Synthons for Molecular Construction—Oligofunctionalization of Cobalt Bis(1,2-dicarbollide) on Boron and Carbon Atoms with Extendable Ligands

**DOI:** 10.3390/molecules28104118

**Published:** 2023-05-16

**Authors:** Krzysztof Śmiałkowski, Carla Sardo, Zbigniew J. Leśnikowski

**Affiliations:** 1Laboratory of Medicinal Chemistry, Institute of Medical Biology Polish Academy of Sciences, Lodowa 106, 93-232 Lodz, Poland; ksmialkowski@cbm.pan.pl (K.Ś.); csardo@unisa.it (C.S.); 2Lodz Institutes of the Polish Academy of Science, The Bio-Med-Chem Doctoral School, University of Lodz, 90-237 Lodz, Poland; 3Department of Pharmacy, University of Salerno, Via Giovanni Paolo II, 132, 84084 Fisciano, SA, Italy

**Keywords:** metallacarboranes, cobalt bis(1,2-dicarbollide), oligofunctionalization, alkylation, stereochemistry

## Abstract

The exploitation of metallacarboranes’ potential in various fields of research and practical applications requires the availability of convenient and versatile methods for their functionalization with various functional moieties and/or linkers of different types and lengths. Herein, we report a study on cobalt bis(1,2-dicarbollide) functionalization at 8,8′-boron atoms with different hetero-bifunctional moieties possessing a protected hydroxyl function allowing further modification after deprotection. Moreover, an approach to the synthesis of three and four functionalized metallacarboranes, at boron and carbon atoms simultaneously via additional functionalization at carbon to obtain derivatives carrying three or four rationally oriented and distinct reactive surfaces, is described.

## 1. Introduction

Boron clusters, a class of polyhedral caged compounds, are playing an increasingly prominent role in the development of a broad range of technologies in fields such as material science [1,2] and medicinal chemistry [3,4,5]. They also find applications as labels for nucleic acid fragments [6,7,8]. Conjugates of DNA oligonucleotides and functionalized boron clusters have recently been proposed as building blocks for the construction of new nanomaterials for biomedical applications [9,10].

The ability of a boron cluster’s type of dicarbaborate anion (*nido*-7,8-C_2_B_9_H_11_) to coordinate a wide spectrum of metal ions, such as Fe, Co, Cr, Ta, Mo, W, V, and Nb, and form metal complexes, namely, metallacarboranes, extends the range of its prospective applications [11,12,13,14]. Among them, the most widely used metallocarborane is cobalt bis(1,2-dicarbolide), which is a sandwich of two [C_2_B_9_H_11_]^2−^ (dicarbollide) units with a cobalt ion in the center of the complex structure.

The broad technological potential of metallacarboranes requires access to a diverse array of functionalities (reactive functional groups, alkyl chains, spacers or polymers of various lengths, pharmacologically active species, etc.). Furthermore, it is necessary for more than one of these kinds of functionalities to coexist in the same metallacarborane structure to produce hetero-functionalized, tailor-made metallacarborane derivatives. The possibility to arrange these functionalities in a specific spatial orientation with respect to the topology of the 3D cluster core facilitates the achievement of the desired covalent modification, providing atomic-level precision and allowing for the control of size and surface composition. Finally, a highly relevant aspect regarding molecular design is the possibility of distancing the metallacarborane cage from the introduced functional group and the prospect of their further modification and/or extension. 

The successful use of functionalized 1,2-dicarbadodecaborane as a core unit in the synthesis of building blocks containing a boron cluster and DNA for the construction of functional nanoparticles carrying therapeutic nucleic acids [9,10] prompted us to extend this technology towards metallacarboranes. Due to the different shapes of metallacarboranes, it may be possible to obtain nanoparticles with a different topology than that of 1,2-dicarbadodecaborane, which, in turn, can affect their biological properties. The ability of boron clusters to function as membrane carriers for a broad range of cargo molecules, thereby facilitating therapeutic nucleic acid cellular uptake, constitutes an additional potential advantage of metallacarborane-containing DNA nanoparticles [15,16,17,18].

## 2. Results

Herein, the results of a study on the oligofunctionalization of cobalt bis(1,2-dicarbollide) at boron atoms 8 and 8′ and 1,1′ and/or 2,2′ at carbon atoms with extendable ligands are described. Ligands such as hydroxyalkyls, which possess a hydroxy function separated from the metallacarborane core by spacers, were used. They offer the possibility of maximizing the distance of functional moieties from each other and the cluster core to avoid the influence of metallacarborane (known as the “metallacarborane effect”) [19] on the chemical properties of these moieties. Analogous (but unprotected and substituted at cage carbon atoms only) alkylhydroxy derivatives of cobalt bis(1,2-dicarbollide) were described by Grüner and colleagues [20,21]. 

Moreover, derivatives were designed to test the applicability of two types of protections commonly employed in the chemical synthesis of DNA hydroxyl-protecting groups, namely, trityl and alkylsilyl protections, to allow for chemo-selection in a subsequent chemical manipulation. These results are complemented by studies on the further functionalization of the metallocarborane substituted on the 8,8′ boron atoms via substitution on the 1,1′ and 2,2′ carbon atoms of the carborane ligands.

### 2.1. Functionalization at 8 and 8′ Boron Atoms via Direct Alkylation of Hydroxy Groups in 8,8′-Dihydroxy-bis(1,2-dicarbollide)-3-cobalt(1-)ate (***2***)

The functionalization procedure began with converting cobalt bis(1,2-dicarbollide) (**1**) into easily available 8,8′-dihydroxy-bis(1,2-dicarbollido)-3-cobalt(1-)ate (**2**) in a reaction with 80% aqueous sulfuric acid [22]. The substitution reaction proceeded selectively at the 8 and 8′ boron atoms, which are the ones with maximum electron density [23]. 

As the alkylating agents used for compound **2** were 4-(trityloxy)butyl-4-methylbenzenesulfonate (**3**) [24] or (3-bromopropoxy)-*tert-*butyldimethylsilane (**4**), which differ in terms of leaving groups (tosyl or bromine) and hydroxyl group protection (trityl, -CPh_3_ or *tert*-butyldimethylsilyl, TBDMS), NaH was used as a base to activate the hydroxyl groups in **2** (Figure 1). 

For the reactions involving both tosylate **3** and bromide **4**, even if an excessive amount of NaH was used, the complete alkylation of both hydroxyl groups was not achieved after 24 h, yielding a mixture of mono-substitution products **5** or **6** (minor products) and bis-substitution products **7** or **8** (major products). Both mono- and bis-alkylated products are easily separable via column chromatography conducted on silica gel using a gradient of methanol or acetonitrile in chloroform as the eluting solvent system. 

The final yield of trityl-protected **7** after purification was found to vary considerably, ranging from 30 to 60%, although the yield of conversion detected in the crude reaction mixture was high, with a bis-functionalized derivative as the major product formed. This variability in the recovery of the tritylated products **5** and **7** can be ascribed to the relative instability of the trityl protection in **5** and **7**. A reason for this instability could be the known Lewis acidity of the boron cluster cage [25] and the acid lability of trityl protection. Interestingly, it seems that one of the factors influencing this effect may be the distance of the trityl protection from the metallocarborane core because in compounds **15**, **16,** and **21**, which contain longer linkers, the instability of the trityl groups does not appear to be a problem.

Compounds **6** and **8** containing silyl protections are reasonably stable anions that can be purified and separated via column chromatography on silica gel. TBDMS protecting groups were quantitatively removed from terminal hydroxyls via treatment with TBAF in THF at room temperature, as demonstrated for **8** (Figure 2). This treatment led to a counterion exchange in **9,** which was demonstrated using NMR analysis. 

As indicated via ^1^H NMR (Appendix A), alkylated compounds **5–8** were isolated in the form of tetraalkylammonium (CH_3_)_2_NR_2_ salt, where R_2_ is -(CH_2_)_4_OCPh_3_ or -(CH_2_)_3_OSi(CH_3_)_2_C(CH_3_)_3_ (Figure 1). The unexpected structure of the tetraalkylammonium counterions is most probably due to the in situ formation of sodium dimethylamine as a result of a side reaction involving the reduction of DMF, used as a solvent by NaH, and its reaction with the alkylating agent [26]. The modest nucleophilicity of the hydroxyls on one side and the erosion of reagents because of this competing reaction with solvent on the other side account for the use of an excess of reagents to achieve good yields. These undesirable properties of the pair of NaH and DMF are also responsible for the incomplete alkylation of both hydroxyl groups in **2** mentioned above under the conditions employed, despite the other advantages of DMF solvent.

Next, we attempted to functionalize the carbon atoms of the bisubstituted on the boron atoms compound **8** (Figure 3). The reaction was carried out in anhydrous DME at a temperature ranging from −70 °C to room temperature according to a typical procedure. First, to a solution of **8** in DME, nBuLi in hexane and then ethylene oxide solution in THF were added. After 24 h, the reaction was quenched, which, after a standard workup, yielded the mono-substitution product **10** formed as the minor product and the bis-substitution product **11** formed as the major one. Although moderate amounts of the target compounds can be obtained using this approach, the synthetic yields are generally not high.

We hypothesize that the reasons for the limited susceptibility of the type **8** compounds to functionalization on carbon atoms may include the electron-donating effect of boron cluster ligands; steric hindrance due to the presence of large substituents at the 8 and 8′ positions, thereby hampering the electrophile’s access to the activated carbon atoms; the formation of intramolecular hydrogen bonds between the oxygen atoms of the substituents at the 8,8′ position and acidic carborane C-H groups (Figure 1) analogously to C-H⋅⋅⋅X-B hydrogen bonds [27,28]; and rotations of 1,2-dicarbollide ligands around an axis, thereby decreasing the susceptibility of C-H groups to activation and alkylation. Consequently, we decided to test a different approach based on derivatives of cobalt bis(1,2-dicarbollide) with arrested rotation and containing a phosphorothioate bridging moiety, namely, **13**.

### 2.2. Functionalization at Boron Atoms 8 and 8′ via S-Alkylation of 8,8′-O,O-[Cobalt bis(1,2-dicarbollide)]phosphorothioate (***13***)

#### 2.2.1. Synthesis of 8,8′-Bridged 8,8′-O,O-[Cobalt bis(1,2-dicarbollide)]phosphorothioate (**13**)

The target compound **13** was obtained through a simple, two-step procedure (Figure 4). In the first step, the 8,8′-dihydroxy-derivative **2** was converted into 8,8′-bridged 8,8′-O,O-[cobalt bis(1,2-dicarbollide)] H-phosphonate acid ester (**12**) via a reaction with tris(1H-imidazol-1-yl)phosphine [29] and the in situ hydrolysis of the resultant imidazolide. 

In the second step, H-phosphonate acid ester **12** was dissolved in anhydrous methanol; then, elemental sulfur S_8_ was added. A strong organic base, 1,8-diazabicyclo(5.4.0)undec-7-en (DBU), was added to the resultant suspension, and the reaction was left for 96 h at room temperature with stirring. After the evaporation of the methanol, the residue containing the crude product was purified via silica gel column chromatography using a gradient of acetonitrile in chloroform as the eluting solvent system. 

#### 2.2.2. S-Alkylation of 8,8′-Bridged 8,8′-O,O-[Cobalt bis(1,2-dicarbollide)]phosphorothioate (**13**) with Linear and Branched Alkylating Agents

Sulfur is a larger atom than oxygen, rendering its electrons more polarizable and the atom itself more nucleophilic. The alkylation of the sulfur atom of phosphorothioates is a viable method for the synthesis of their S-alkylated derivatives. This method takes advantage of the excellent nucleophilic properties of sulfur and is commonly used in organophosphorus chemistry [30].

Using these advantageous properties of phosphorothioates, we attached both linear **13** and **14** or branched **20** ligands containing hydroxyl functions protected by a trityl group to the metallocarborane derivative **13**. The alkylation reaction proceeded smoothly, and the yields of the isolated alkylated products **15**, **16**, and **21** were high (Figure 5). 

It is worth noting that derivatives of known 8,8′-bridged 8,8′-O,O-[cobalt bis(1,2-dicarbollide)]phosphate [22], a counterpart of **13** containing an 8,8′-O,O-phosphate bridge instead of a phosphorothioate one, have not yet been described. The expected low nucleophilicity of the phosphorus center resulting from the metallocarborane effect [19] and the lower nucleophilicity of oxygen compared to sulfur atom can be one of the reasons for this.

Consequently, a study on the functionalization of **13′**s carbon atoms was undertaken. For this purpose, a method for the synthesis of a branched alkylating agent (**20**), in this case, a glycerol derivative, was first developed. In the first step, bis(trityloxy)propan-2-ol (**18**) was synthesized according to a method reported in the literature [31]. Next, **18** was reacted with 1,4-bis(*p*-toluenesulfonyloxy)butane (**19**), yielding a branched alkylating agent with an elongated linker (**20**) (Figure 6). 

The functionalization of **21** on carbon atoms was achieved via the activation of cage C-H groups with nBuLi followed by treatment with ethylene oxide. As expected, carrying out the synthesis of **22** and **23** required overcoming the low susceptibility of intermediate **21** to substitution on carbon atoms. However, mono- and disubstituted derivatives **22** and **23** were obtained with yields that enabled their full characterization and use for further chemical manipulations. The accessibility and high yields offered by the synthesis of intermediates **13** and **21** allow for the easy scaling-up of the synthesis of **22** and **23** (if needed).

## 3. Discussion

In the literature, the description of metallacarborane building blocks with hydroxyl or other functional groups separated from a cluster cage by an alkyl spacer seems limited. In this contribution, we report the development of methods for the functionalization of cobalt bis(1,2-dicarbollide) (**1**) with hetero-bifunctional derivatives of silyl- or trityl-protected alcohols attached directly to boron atoms and trityl-protected branched alcohols attached to metallacarborane through a phosphorothioate bridge. An approach to the synthesis of oligofunctionalized metallacarboranes via carbon deprotonation through nBuLi to obtain derivatives carrying substituents at both boron and carbon atoms is also described. 

After a long period of discussion, the aromatic, three-dimensional (3D) nature of boron clusters is now widely accepted [32,33]. As in the case of aromatic organic molecules, the attachment of a substituent to one of the atoms of a 3D aromatic boron cluster system changes the distribution of the electron density of the entire molecule and affects the properties of other reactive centers. This makes the oligofunctionalization of metallacarboranes, especially when performed simultaneously on boron and carbon atoms, a particular challenge. Additionally, the complicated stereochemistry of substituted metallacarboranes owing to the presence of various types of chirality in the same molecule further compounds this challenge.

One of the practical manifestations of an effect of synchronized changes in electron density within the whole metallacarborane molecule on the properties of reactive centers is the inclination for the formation of disubstituted derivatives on boron atoms 8 and 8′ and carbon atoms 1,2 and 1′,2′. This is illustrated by the preferential formation of disubstituted derivatives compared to monosubstituted ones in the case of **7** vs. **5**; **8** vs. **6**; **11** vs. **10**; and **23** vs. **22**. This is clearly due to the activation of a second nucleophilic center, such as B-OH or C-H groups, by a strong base after a previous substitution at the first center. However, this does not change the fact that the overall alkylation efficiency of the C-H groups in the derivatives of **1** already substituted on 8 and 8” boron atoms is low.

For example, the monoalkylation derivatives of 8,8′-dihydroxy-bis(1,2-dicarbollide)-3-cobalt(1-)ate (**2**) and bisalkylation derivatives were isolated in a ratio of 1:25 for **5** and **7** and 1:10 for **6** and **8**. The mono- and bis-substituted at carbon atoms derivatives of the bis-substituted at boron atoms of compound **8** were isolated in a 1:3 ratio for **10** and **11**. The same trend, though less pronounced, was observed for the boron and carbon functionalized derivatives **22** and **23** with arrested rotation, for which the ratio of mono-substitution on carbon bis-substitution was 1:1.2. 

Another property significantly influencing the chemistry of oligofunctionalized metallacarboranes is their stereochemistry. The phenomenon of boron cluster chirality was recognized early [34,35,36]; however, this did not initially arouse much interest. The increasing use of boron clusters in the field of new materials, nanotechnology, and medical chemistry renders the stereochemistry of boron clusters increasingly important.

Strangely, though the most symmetric species in nature is the B_12_H_12_^−^ ion, relatively minor changes in the boron cluster structure might render its basic framework dissymmetric enough to provoke chirality [37,38]. This is particularly evident in the case of oligofunctionalized metallacarboranes. The derivatives of type **22** and **23** are extreme examples containing various sources of chirality such as a chiral center at the phosphorus atom, axial and planar chirality due to bending of the metallacarborane molecule, and the existence of a number of isomers due to substitution at carbon atoms in addition to boron substitution.

A comparison of the ^11^B-NMR spectra for compounds **13**, **15**, **16**, **21**, **22**, and **23** with arrested rotation as well as their ^31^P-NMR spectra clearly shows changes in the number of isomers of the individual products depending on the type of substitution. Compounds **13**, **15**, **16,** and **21** show a singlet at δ of about 23 ppm attributed to 2B(8,8′) in the ^11^B-NMR spectrum and a singlet at 48.62 ppm for **13** of about 16 ppm for compounds **15**, **16**, and **21** in the ^31^P-NMR spectrum; these findings are consistent with the relative symmetry of the complexes. 

On the contrary, the ^11^B- and ^31^P-NMR spectra of mono- and bisalkylated at carbon-atoms derivatives **22** and **23** show a dramatic change in symmetry due to the many possible combinations of substitutions on carbon atoms and the formation of a center of chirality on the phosphorus atom in monosubstituted **22** and in some isomers of bis-substituted **23**. In consequence, four signals at 25.37, 24.65, 23.69, and 22.93 ppm with an integral intensity ratio of 2:2:1:1 corresponding to B8,8′ for **22** and five signals at 25.35, 24.40, 23.64, 22.80, and 22.23 ppm with an approximate integral intensity ratio of 1:2:1.5:1:1 for **23** were observed. A similar effect was observed in the ^31^P-NMR spectra, with seven signals at 15.10, 14.94, 14.57, 14.42, 14.11, 13.74, and 13.49 ppm for **22** and five signals at 14.99, 14.39, 14.16, 14.01, and 13.38 ppm for **23**, thus reflecting the asymmetry of these compounds and formations of isomers.

The sensitivity of standard chromatographic techniques is insufficient for distinguishing these isomers and only allows for the separation of mono- and disubstituted derivatives on carbon atoms, which have already been substituted on boron atoms in the case of **10** and **11** and **22** and **23**. A complete or partial separation of the isomers is probably attainable using HPLC; however, with regard to further practical applications of these derivatives, it is not necessary at the present stage.

## 4. Materials and Methods

### 4.1. Materials

All solvents were purchased in the highest available quality required for their application in this study. The reactions requiring anhydrous conditions were carried out under argon atmosphere using anhydrous solvents treated with activated molecular sieves for at least 24 h. Molecular sieves (4Å and 3Å) were purchased from Alfa Aesar (Karlsruhe, Germany) and heat-activated under vacuum before use. Triethylamine, trityl chloride, toluene-sulfonyl chloride, NaHCO_3_, Na_2_SO_4_, 1,4-butanediol, P_2_O_5_, sodium hydride, n-butyllithium (1.6 M solution in hexane), (3-bromopropoxy)-*tert*-butyldimethylsilane, (**4**) and ethylene oxide (2.9–3.1 M solution in THF) were purchased from Sigma Aldrich (Steinheim, Germany). Cobalt bis(1,2-dicarbollide) was purchased from Katchem Ltd., Prague, Czech Republic. 

### 4.2. Methods

**Chromatography.** Ultramate 3000 HPLC system (DIONEX, Sunnyvale, USA). equipped with a photodiode array detector (with fixed wavelengths of 210, 270, 310, and 330 nm) was used to determine the purity of products. The method consisted of a gradient elution from 20% to 90% aqueous acetonitrile using a Hypersil Gold (5μm particle size) reverse-phase column at 25 °C. HPLC data were acquired and processed using Chromeleon 6.8 software (DIONEX, Sunnyvale, CA, USA). Chromatography for purification of products was performed on a 230–400 mesh silica-gel Sigma Aldrich (Steinheim, Germany) filled glass column. Analytical TLC was performed on F254 silica gel plates purchased from Sigma Aldrich. Compounds were visualized using UV light (254 nm) and/or via staining with 0.05% *w*/*v* palladium chloride solution in MeOH/HCl. 

**NMR spectroscopy.** ^1^H, ^11^B, ^11^B{^1^H}, ^13^C{^1^H}, ^31^P, and ^31^P{^1^H} NMR spectra were recorded with a Bruker Advance III 600 MHz spectrometer.

**UV-Vis spectrophotometry** and **line-fitting analysis.** Measurements were performed using a Jasco V-750 UV spectrophotometer at room temperature in acetonitrile.

**MS** and **FT-IR.** MALDI-TOF MS spectra were recorded using a Voyager–Elite mass spectrometer (PerSeptive Biosystems) with 3-hydroxypicolinic acid (HPA) as the matrix. ESI MS mass spectra were recorded using Agilent 6546 LC/Q-TOF (Santa Clara, Kalifornia, United States). Negative ions were detected. Infrared absorption spectra were recorded using a Nicolet 6700 FT-IR spectrometer (Thermo Scientific) equipped with a Smart orbit diamond Attenuated Total Reflectance (ATR) accessory. Samples to be analyzed were placed on a diamond ATR element in a solid form or through casting from CH_2_Cl_2_ solution. Data were acquired and processed using Omnic 8.1 software (Thermo Scientific, Waltham, CA, USA).

*8,8′-dihydroxy-bis(1,2-dicarbollido)-3-cobalt(1-)ate HNEt_3_* (**2**) was synthesized according to the procedure reported by Plešek et al. [22]. 

*4-Trityloxybutyl 4-methylbenzenesulfonate* (**3**) was obtained as described in [39]. 

*3-Bromo-1-trityloxypropane* (**14**) was synthesized as described previously [10].

*1,3-bis(trityloxy)propan-2-ol* (**18**) was synthesized according to a procedure in the literature [31].

*1,4-bis-(4-methylbenzenesulfonate)butane* (**19**) was synthesized according to a modified procedure from the literature [40].

*Synthesis of 3,3′-Co{[(8-O(CH_2_)_4_OCPh_3_]-1,2-C_2_B_9_H_10_}(8′-OH-1,2-C_2_B_9_H_10_) (CH_3_)_2_N[(CH_2_)_4_OPh_3_]_2_ (***5***) and 3,3′-Co[(8-O(CH2)_4_OCPh_3_-1,2-C_2_B_9_H_10_)]_2_ (CH_3_)_2_N[(CH_2_)_4_Oph_3_]_2_ (***7***).* Compound **2** (50 mg, 0.10 mmol) was dissolved in 0.5 mL of anhydrous dimethoxyethane (DME) and added to 60% NaH in mineral oil dispersion (4.4 mg, 0.10 mmol NaH) under argon atmosphere. The reaction mixture was stirred for 2 h at room temperature. After this time has elapsed, the solvent was evaporated under reduced pressure, and the resultant solid residue was dissolved in 0.5 mL of anhydrous dimethylformamide (DMF) and added to a second aliquot of 60% NaH in mineral oil dispersion (26.2 mg, 0.65 mmol NaH). After sitting for 2 h at room temperature, the mixture was dropped into a solution of **3** (318.8 mg, 0.65 mmol) in 0.5 mL of anhydrous DMF; then, the reaction mixture was heated at 80 °C (oil bath temperature) for 24 h. The post-reaction mixture was concentrated; then, 3 mL of H_2_O was added. The resulting precipitate was washed with several aliquots of H_2_O. Then, the solid was dissolved in 5 mL of CH_2_Cl_2_, and the solution was dried with anhydrous Na_2_SO_4,_ filtered, and concentrated. Repeated column chromatography on silica gel using a gradient of CH_3_OH in CH_3_Cl (0 to 20%) as an eluting solvent system yielded **5** and **7** as the first and second bands, respectively. Fractions containing compound **7** were collected, evaporated to dryness, and then crystallized from hexane, furnishing 96% pure product as determined by HPLC. Monosubstituted compound **5** was obtained as a red oil. 

(**5**): **Yield**: 7 mg (5%); **TLC** (CHCl_3_:MeOH 4:1); **R_f_**: 0.53; **MALDI-MS** (*m*/*z*): 670.5 (calc. for C_27_B_18_H_44_O_3_Co_1_: 670.17). Since excessively small quantities of this product were obtained, it was not further analyzed via NMR.

(**7**): **Yield**: 180 mg (62%); **TLC** (CHCl_3_:MeOH 4:1); **R_f_**: 0.64; **^1^H NMR** (500 MHz, CD_3_CN) δ: 7.43 (m, 24H, *H_arom_*), 7.32 (m, 24H, *H_arom_*), 7.25 (m, 12H, *H_arom_*), 4.18 (s, 4H, *CH_carborane_*), 3.31 (t, *J* = 6.2 Hz, 4H, BO*CH_2_*CH_2_CH_2_CH_2_OTr), 3.07 (m, 8H, *CH_2_*OTr), 2.99 (t, *J* = 6.4 Hz, 4H, N*CH_2_*CH_2_CH_2_CH_2_OTr), 2.86 (s, 6H, N*(CH_3_)_2_*), 1.68 (m, 4H, NCH_2_*CH_2_*CH_2_CH_2_OTr), 1.60 (m, 8H, *CH_2_*CH_2_OTr), and 1.48 (m, 4H, BOCH_2_*CH_2_*CH_2_CH_2_OTr); **^13^C{^1^H} NMR** (126 MHz, CD_3_CN) δ: 145.53 and 145.18 (12C, *aromatic_trityl_*); 129.44 and 129.41 (24C, *aromatic_trityl_*); 129.37 and 128.83 (24C, *aromatic_trityl_*); 128.21 (12C, *aromatic_trityl_*); 87.43 and 87.06 (4C, *C(Ph)_3_*), 69.43 (2C, BO*CH_2_*CH_2_CH_2_CH_2_OTr), 64.30 (2C, BOCH_2_CH_2_CH_2_*CH_2_*OTr), 63.30 (2C, NCH_2_CH_2_CH_2_*CH_2_*OTr), 62.33 (2C, N*CH_2_*CH_2_CH_2_CH_2_OTr), 51.56 (4C, *CH_carb_*), 30.39, 29.68 (4C, *CH_2_*CH_2_OTr), 27.51, 27.13 (2C, BOCH_2_*CH_2_*CH_2_CH_2_OTr), and 20.29 (NCH*_2_CH_2_*CH_2_CH_2_OTr); **^11^B{^1^H} NMR** (160 MHz, CD_3_CN) δ: 20.63 (s, 2B, B^8,8′^), −3.56 (s, 2B, B^10,10′^), −7.52 (s, 4B, B^4,4′,7,7′^), −9.03 (s, 4B, B^9,9′,12,12′^), −20.53 (s 4B, B^5,5′,11,11′^), and −28.38 (s, 2B, B^6,6′^) **^11^B NMR** (160 MHz, CD_3_CN) δ: 20.66 (s, 2B, B^8,8′^), −3.59 (d, 2B, B^10,10′^), −8.22 (m, 8B, B^9,9′,12,12′,4,4′,7,7′^), −20.48 (d, 2B, B^5,5′,11,11′^), and −28.09 (d, 2B, B^6,6′^); **FT-IR** (cm^−1^): 3083.53, 3055.16, 3029.62 (ν C-H aromatic, (C-H_carb_); 2927.18 (ν C-H_asym_, CH_2_); 2867.43 (ν C-H, CH_2_O, ν C-H_sym_, CH_2_); 2543.23 (ν B-H); 1596.06, 1488.72, 1447.49 (ν C=C); 1152.23; 745.10 and 693.37 (δ C-H aromatic); 703.89 (ν B-B); **MALDI-MS** (*m*/*z*): found 984.2 (calc. for C_50_B_18_H_66_O_4_Co_1_: 984.59).

*Synthesis of 3,3′-Co [8-O(CH_2_)_3_OTBDMS]-1,2-C_2_B_9_H_10_)(8′-OH-1,2-C_2_B_9_H_10_) (CH_3_)_2_N[(CH_2_)_3_OTBDMS)]_2_ (***6***) and 3,3′-Co[8-O(CH_2_)_3_OTBDMS-1,2-C_2_B_9_H_10_)]_2_ (CH_3_)_2_N[(CH_2_)_3_OTBDMS)]_2_ (***8***)*. Compound **2** (300 mg, 0.65 mmol) was dissolved in 3 mL of anhydrous DME under argon atmosphere and added to 60% NaH in mineral oil dispersion (26.2 mg, 0.65 mmol NaH). The reaction mixture was stirred for 2 h at room temperature. Subsequently, the solvent was evaporated, and the solid residue was dissolved in 3 mL of anhydrous DMF and added to a second portion of 60% NaH in mineral oil dispersion (157 mg, 3.93 mmol NaH). After sitting for 2 h at room temperature, the mixture was heated to 60 °C (oil bath temperature) and 911 μL of (3-bromopropoxy)(*tert*-butyl)dimethylsilane (**4**) (3.93 mmol) was added dropwise. After 22 h, an additional quantity of **4** (600 μL, 2.59 mmol) was added, and the mixture was stirred for next 24 h at 60 °C. A white solid was formed. The post-reaction mixture was then filtered through syringe filter (5 μm, PTFE, Carl Roth), and DMF was evaporated under reduced pressure. The solid residue was treated with 2 mL of CHCl_3_ and filtered; then, the the filtrate was concentrated and loaded on a silica gel column for separation of products. First, 100% CHCl_3_ and then 5% and 10% CH_3_CN in CHCl_3_ were used as eluting solvent systems. Compound **8** was isolated as first fraction and obtained in the form of orange crystals after solvent evaporation; its purity was above 95% as determined via HPLC. The second fraction containing **6** was obtained as red oil after solvent evaporation. Both products **6** and **8** were isolated as N,N-bis[(3-(*tert*-butyldimethylsilyloxypropyl)]-N,N-dimethyl ammonium salts [26].

(**6**): **Yield**: 60 mg (10%); **TLC** (CHCl_3_:CH_3_CN 3:1) **R_f_**: 0.53; **^1^H NMR** (600 MHz, CD_3_CN) δ: 6.05 (s, 1H, *OH*), 3.73 (t, *J* = 6.5 Hz, 2H, BOCH_2_CH_2_*CH_2_*OSi), 3.69 (m, 4H, N*CH_2_*CH_2_CH_2_OSi), 3.65 (m, 2H, BO*CH_2_*CH_2_CH_2_OSi), 3.56 (s, 2H, *CH_carb_*), 3.45 (s, 2H, *CH_carb_*), 3.27 (m, 4H, NCH_2_CH_2_*CH_2_*OSi), 2.97 (s, 6H, N*(CH_3_*)_2_), 1.88 (m, 4H, NCH_2_*CH_2_*CH_2_OSi), 1.74 (m, *J* = 6.4 Hz, 2H, BOCH_2_*CH_2_*CH_2_OSi), 0.90 (s, 9H, BOCH_2_CH_2_CH_2_OSi(CH_3_)_2_C(*CH_3_*)_3_), 0.89 (s, 18H, NCH_2_CH_2_CH_2_OSi(CH_3_)_2_C(*CH_3_*)_3_), 0.07 (s, 6H, BOCH_2_CH_2_CH_2_OSi(*CH_3_*)_2_C(CH_3_)_3_), and 0.05 (s, 12H, NCH_2_CH_2_CH_2_OSi(*CH_3_*)_2_C(CH_3_)_3_); **^13^C{^1^H} NMR** (126 MHz, CD_3_CN) δ: 67.05 (1C, BO*CH_2_*CH_2_CH_2_OSi), 62.56 (2C, NCH_2_CH_2_*CH_2_*OSi), 61.23 (1C, BOCH_2_CH_2_*CH_2_*OSi), 60.17 (2C, N*CH_2_*CH_2_CH_2_OSi), 52.03 (2C, N(CH_3_)_2_), 46.23 (2C, CH_carb_), 45.51 (2C, CH_carb_), 35.92 (1C, BOCH_2_*CH_2_*CH_2_OSi), 26.45 (2C, NCH_2_*CH_2_*CH_2_OSi), 26.27 (6C, OSi(CH_3_)_2_C*(CH_3_)_3_*), 26.11 (3C, OSi(CH_3_)_2_C*(CH_3_)*_3_), 18.81 (1C, OSi(CH_3_)_2_*C*(CH_3_)_3_), 18.72 (2C, OSi(CH_3_)_2_*C*(*CH_3_)_3_*), −5.10 (2C, OSi*(CH_3_)*_2_C(CH_3_)_3_), and −5.40 (4C, OSi(*CH_3_)*_2_C(CH_3_)_3_); **^11^B{^1^H} NMR** (193 MHz, CD_3_CN) δ: 27.16 (s, 1B, B^8′^), 25.10 (s, 1B, B^8^), −5.02 to −9.09 (10B, overlapped B^9,9′,10,10′12,12′,4,4′,7,7′^), −20.10 to −20.69 (4B, B^5,5′,11,11′^), and −29.18 to −30.16 (2B, B^6,6′^); **^11^B NMR** (160 MHz, CD_3_CN) δ: 27.16 (s, 1B, B^8′^), 25.09 (s, 1B, B^8^), −3.22 to −9.40 (10B, overlapped B^9,9′,10,10′12,12′,4,4′,7,7^) −19.72 to −21.11 (4B, B^5,5′,11,11′^), and −28.81 to −30.49 (2B, B^6,6′^); **FT-IR** (cm^−1^): 3278.64, 2952.00, 2927.58, 2855.64, 2539.99, 1470.65, 1387.53, 1360.69, 1252.66, 1156.76, 1093.25, 1005.54, 972.16, 938.48, 880.24, 832.61, 774.98, 718.29, and 661.26; **ESI-MS** (*m*/*z*): found 530.39 and 472.35 [M-*t*Butyl]^−^ (calc. for C_13_B_18_H_43_Si_1_O_3_Co_1_: 529.09).

(**8**): **Yield**: 572 mg (80%); **TLC** (CHCl_3_:CH_3_CN 3:1) **R_f_** 0.70; **^1^H NMR** (500 MHz, CD_3_CN) δ: 4.18 (s, 4H, *CH_carb_*), 3.69 (m, 4H, BOCH_2_CH_2_*CH_2_*OSi), 3.64 (t, *J* = 6.4 Hz, 4H, N*CH_2_*CH_2_CH_2_OSi), 3.38 (t, *J* = 6.0 Hz, 4H, and BO*CH_2_*CH_2_CH_2_OSi), 3.27 (m, 4H, NCH_2_CH_2_*CH_2_*OSi), 2.97 (s, 6H, N(*CH_3_)_2_*), 1.88 (m, 4H, NCH_2_*CH_2_*CH_2_OSi), 1.58 (m, *J* = 6.2 Hz, 4H, BOCH_2_*CH_2_*CH_2_OSi), 0.90 (s, 18H, OSi(CH_3_)_2_C*(CH_3_)_3_*), 0.88 (s, 18H, OSi(CH_3_)_2_C*(CH_3_)_3_*), 0.07 (s, 12H, OSi*(CH_3_)_2_C(CH_3_)_3_), and 0.04 (s, 12H, OSi(CH_3_)_2_C(CH_3_)_3_); **^13^C NMR** (126 MHz, CD_3_CN)* δ: 66.18 (2C, BOCH_2_CH_2_*CH_2_*OSi), 62.54 (2C, NCH_2_CH_2_*CH_2_*OSi), 60.95 (2C, BO*CH_2_*CH_2_CH_2_OSi), 60.17 (2C, N*CH_2_*CH_2_CH_2_OSi), 52.06 (2C, N*(CH_3_)_2_*), 51.60 (4C, *CH_carb_*), 35.97 (2C, BOCH_2_*CH_2_*CH_2_OSi), 26.46 (2C, NCH_2_*CH_2_*CH_2_OSi), 26.28 (6C, OSi(CH_3_)_2_C*(CH_3_)_3_*), 26.12 (6C, OSi(CH_3_)_2_C*(CH_3_)_3_* 18.83 (2C, OSi(CH_3_)_2_C(CH_3_)_3_), 18.72 (2C, OSi(CH_3_)_2_C(CH_3_)_3_), −5.09 (4C, OSi(*CH_3_)*_2_C(CH_3_)_3_), and −5.39 (4C, OSi(*CH_3_)*_2_C(CH_3_)_3_); **^11^B{^1^H} NMR** (160 MHz, CD_3_CN) δ: 20.61 (s, 2B, B^8,8′^), −3.61 (s, 2B, B^10,10′^), −7.45 (s, 4B, B^4,4′,7,7′^), −9.03 (s, 4B, B^9,9′,12,12′^), −20.51 (s, 4B, B^5,5′,11,11′^), and −28.38 (s, 2B, B^6,6′^); **^11^B NMR** (160 MHz, CD_3_CN) δ: 20.76 (s, 2B, B^8,8′^), −3.59 (d, 2B, B^10,10′^), −7.18 to −9.37 (m, 8B, overlapped B^4,4′,7,7′,9,9′,12,12′^), −20.48 (d, 4B, B^5,5′,11,11′^), and −28.02 (d, 2B, B^6,6′^); **FT-IR** (cm^−1^): 3048.6 (C-H_carb_); 2949.4 (ν C-H_asym_, CH_3_); 2927.4 (ν C-H_asym_, CH_2_), 2891.6 and 2884.2 (ν C-H_sym_ CH_3_); 2856.2 (ν C-H, CH_2_O, ν C-H_sym_, CH_2_); 2605.2 and 2550.6 (ν B-H); 1470.8 (δ C-H_sym_, CH_2_) and 1436.0; 1386.2; 1359.5; 1251.2 (Si-CH_3_) and 1161.9 (Si-O-C); 1093.1 (ν Si-O); 1019.3; 1006.3; 975.2; 955.3; 942.7; 881.; 831.8 and 772.4 (ν Si-C); 710.9; 661.0; **MALDI-MS** (*m*/*z*): found 700.5 (calc. for C_22_B_18_H_62_CoO_4_Si_2_ 700.43).

*Deprotection of compound* **8** *to obtain [3,3′-Co(8-O(CH_2_)_3_OH-1,2-C_2_B_9_H_10_)_2_] TBA (***9***).* Compound **8** (20 mg, 0.018 mmol) was dissolved in 0.2 mL of tetrahydrofuran (THF). A total of 69 μL of terabutylammonium fluoride (TBAF) (1M solution in THF, 0.069 mmol) was added to the obtained solution, and the reaction mixture was stirred at room temperature overnight. Then, solvent was evaporated under reduced pressure, and the resulting oil was dissolved in CHCl_3_ and applied to the silica gel column prepared in the same solvent. Elution in a gradient 1 to 20% CH_3_CN in CHCl_3_ provided 96% pure **9** (determined by HPLC) as TBA salt. **Yield**: 12.5 mg (95%); **TLC** (CHCl_3_:CH_3_CN 3:1) **R_f_**: 0.47; **^1^H NMR** (500 MHz, CD_3_CN) δ: 4.17 (s, 4H, *CH_carb_*), 3.52 (t, *J* = 6.3 Hz, 4H, BOCH_2_CH_2_*CH_2_*OH), 3.43 (t, *J* = 6.0 Hz, 4H, BO*CH_2_*CH_2_CH_2_OH), 3.10–3.03 (m, 8H, N*CH_2_*CH_2_CH_2_CH_3_), 2.58 (s, 2H, *OH*), 1.59 (m, *J* = 8.2, 3.7 Hz, 12H, BOCH_2_*CH_2_*CH_2_OH, NCH_2_*CH_2_*CH_2_CH_3_), 1.41–1.28 (m, 8H, NCH_2_CH_2_*CH_2_*CH_3_), and 0.96 (t, *J* = 7.4 Hz, 12H, NCH_2_CH_2_CH_2_*CH_3_*); **^13^C NMR** (126 MHz, CD_3_CN) δ: 67.47 (2C, BOCH_2_CH_2_*CH_2_*OH), 60.63 (2C, BO*CH_2_*CH_2_CH_2_OH), 59.23 (4C, N*CH_2_*CH_2_CH_2_CH_3_), 51.39 (4C, *CH_carb_*_orane_), 35.54 (2C, BOCH_2_*CH_2_*CH_2_OH), 24.21 (4C, NCH_2_*CH_2_*CH_2_CH_3_), 20.25 (4C, NCH_2_CH_2_*CH_2_*CH_3_), and 13.72 (4C, NCH_2_CH_2_CH_2_*CH_3_*); **^11^B{^1^H} NMR** (160 MHz, CD_3_CN) δ: 20.95 (s, 2B, B^8,8′^), −3.49 (s, 2B, B^10,10′^), −7.71 (s, 4B B^4,4′,7,7′^), −8.85 (s, 4B, B^9,9′,12,12′^), −20.44 (s, 4B, B^5,5′,11,11′^), and −28.14 (s, 2B, B^6,6′^); **^11^B NMR** (160 MHz, CD_3_CN) δ: 20.97 (s, 2B, B^8,8′^), −3.48 (d, 2B, B^10,10′^), −7.18 to −9.26 (m, 8B, B^4,4′7,7′9,9′,12,12′^), −20.42 (d, 4B, B^5,5′,11,11′^), and −28.19 (d, 2B, B^6,6′^); **FT-IR** (cm^−1^): 3588.76 (ν O-H); 3450.79 (ν N^+^-R); 3052.83 (C-H_carb_); 2962.10 (ν C-H_asym_, CH_2_); 3932.36 and 2873.70 (ν C-H, CH_2_O, ν C-H_sym_, CH_2_); 2529.19 (ν B-H); 1467.75 (ν N^+^-R), 1380.63; 1161.25 (HO-C); 1105.54; 1062.05; 1007.15; 969.00; 945.10; 920.29; 876.90; 789.11; 735.69; 696.07; 664.83; **MALDI-MS** (*m*/*z*): found 472.8 (calc. for C_10_B_18_H_34_O_4_Co_1_ 471.90).

*Synthesis of 3,3′-Co{[8-O(CH_2_)_3_OTBDMS-1-(CH_2_)_2_OH]-1,2-C_2_B_9_H_9_)}[8′-O(CH_2_)_3_OTBDMS-1′,2′-C_2_B_9_H_10_)]^−^ (***10***) and 3,3′-Co[(8-O(CH_2_)_3_OTBDMS-1-(CH_2_)_2_OH-1,2-C_2_B_9_H_9_)]_2_^−^ (***11***).* Compound **8** (50 mg, 0.04 mmol) was dried via co-evaporation with anhydrous benzene and then kept under vacuum, over P_2_O_5_ overnight. Then, it was dissolved in anhydrous DME (1 mL), and the solution was cooled in CO_2_/isopropanol cooling bath. After 15 min, n-BuLi (43 µL, 1.6 M solution in hexane, 1.5 eq) was added, and the reaction mixture was stirred for 10 min. Afterwards, cooling bath was removed, and the mixture was stirred for next 10 min. Then, the reaction mixture was cooled again in cooling bath and another portion of n-BuLi (43 µL) was added. After 15 min, ethylene oxide (60 µL, 2.9–3.1 M solution in THF, 4.5 eq) was added, and the reaction was left overnight in cooling bath. Then, CH_2_Cl_2_ (3 mL) was added to the reaction mixture, the reaction was quenched via addition of water, and the organic solution was washed three times with 5 mL portions of water. Organic layer was separated and dried over MgSO_4_; then, solvents were evaporated. Crude product was purified and mono- and bis-substituted products were separated through silica gel column chromatography using a gradient of MeOH in CH_2_Cl_2_ from 0 to 3% of MeOH.

(**10**): **Yield**: 4.7 mg (9%); **TLC** (MeOH:CH_2_Cl_2_ 1:12.5): **R_f_**: 0.28; **ESI-MS** (*m*/*z*): found 744.55, (calc. for C_24_B_18_H_66_O_5_Si_2_Co_1_ 744.48). Since excessively small quantities of this product were obtained, it was not further analyzed via NMR.

(**11**): **Yield**: 15 mg (27%); **TLC** (MeOH:CH_2_Cl_2_ 1:12.5): **R_f_**: 0.16; **^1^H NMR** (500 MHz, CD_3_CN) δ: 4.32–4.08 (s, 2H, diastereoizomeric *CH_carborane_*), 3.77 to 3.52 (m, 16H, overlapped N*CH_2_*CH_2_CH_2_O, NCH_2_CH_2_*CH_2_*O, BO*CH_2_*CH_2_CH_2_O, BOCH_2_CH_2_*CH_2_*O), 3.52 to 3.39 (m, 4H, HO*CH_2_*CH_2_C_carb_), 3.39 to 3.31 (m, 4H, HOCH_2_*CH_2_*C_carb_), 3.03 (s, 6H, N(*CH_3_)**_2_*), 1.74 (m, 4H, NCH_2_*CH_2_*CH_2_O), 1.65 (m, 4H, BOCH_2_*CH_2_*CH_2_O), 0.90 (s, 18H, NCH_2_CH_2_CH_2_OSi(CH_3_)_2_C(CH_3_)_3_), 0.88 (s, 18H, BOCH_2_CH_2_CH_2_OSi(CH_3_)_2_C(CH_3_)_3_), 0.07 (s, 12H, BOCH_2_CH_2_CH_2_OSi(CH_3_)_2_C(CH_3_)_3_), and 0.05 (s, 12H, NCH_2_CH_2_CH_2_OSi(CH_3_)_2_C(CH_3_)_3_; ^**13**^**C{^1^H} NMR** (126 MHz, CD_3_CN) δ: 67.39 (2C, BO*CH_2_*CH_2_CH_2_O), 66.93 (1C, *CH_carborane_*), 64.82 (1C, *CH_carborane_*), 64.10 (2C, NCH_2_CH_2_*CH_2_*OSi), 62.28 (2C, BOCH_2_CH_2_*CH_2_*O), 61.00 (2C, HO*CH_2_*CH_2_C_carb_), 57.17 (2C, N*CH_2_*CH_2_CH_2_O), 56.24 (2C, N*(CH_3_)*_2_), 53.22 (2C, C_carborane_), 45.17 (2C, HOCH_2_*CH_2_*C_carb_), 36.89 (2C, NCH_2_CH_2_CH_2_O), 27.28 (2C, BOCH_2_*CH_2_*CH_2_O), 26.98 (6C, OSi(CH_3_)_2_C*(CH_3_)*_3_), 26.77 (6C, OSi(CH_3_)_2_C*(CH_3_)*_3_), 19.53 (2C, OSi(CH_3_)_2_*C*(CH_3_)_3_), 19.39 (2C, OSi(CH_3_)_2_*C*(CH_3_)_3_), −4.38 (4C, OSi(*CH_3_)*_2_C(CH_3_)_3_), and −4.73 (4C, OSi*(CH_3_)*_2_C(CH_3_)_3_). **^11^B{^1^H} NMR** (120 MHz, CD_3_CN) δ: 29.65, 25.25, 24.28, 23.33, 21.98 (in ratio: 3:1.5:1:1:10), 31.27 to 19.66 (m, overlapped diastereoizomeric B^8,8′^), −2.63 to −13.75 (m, overlapped diastereoizomeric, B^10,10′,9,9′,12,12′,4,4′,7,7′^), −14.09 to −21.65 (m, overlapped diastereoizomeric B^5,5′,11,11′^), and −22.11 to −29.38 (m, overlapped diastereoizomeric B^6,6′^); **^11^B NMR** (120 MHz, CD_3_CN) δ: 30.85–20.54 (m, overlapped diastereoizomeric B^8,8′^), −2.16 to −13.03 (m, overlapped diastereoizomeric, B^10,10′,9,9′,12,12′,4,4′,7,7′^), −13.91 to −21.45 (m, overlapped diastereoizomeric, B^5,5′,11,11′^), and −21.52 to −27.10 (m, overlapped diastereoizomeric, B^6,6′^); **ESI-MS** (*m*/*z*): found 788.58 (calc. for C_26_B_18_H_70_O_6_Si_2_Co_1_ 788.53).

*Synthesis of 8,8′-bridged [8,8′-O_2_P(O)H-3,3′-Co(1,2-C_2_B_9_H_10_)_2_] HNEt_3_ H-phosphonate (***12***).* Imidazole (0.51 g, 7.5 mmol) was dissolved in minimum ammount of anhydrous acetonitrile. The solvent was evaporated under reduced pressure, and the procedure was repeated twice. After drying under vacuum for 1.5 h, imidazole was redissolved in 16 mL of anhydrous THF and the solution was cooled to −70 °C in a dry ice/isopropanol bath under argon atmosphere. PCl_3_ (210 µL, 2.40 mmol) was added dropwise followed by Et_3_N (1 mL, 7.17 mmol) mixed with 1 mL of anhydrous THF. The entire mixture was stirred at −70 °C for 15 min and then solution of 8,8′-dihydroxy-bis(1,2-dicarbollido)-3-cobalt(1-)ate HNEt_3_ (**2**) (320 mg, 0.69 mmol) in 13 mL of THF was added dropwise. After further 30 min, the reaction mixture was removed from cooling bath and allowed to warm to room temperature. After another 1 h, the reaction was quenched with 30 mL of water and extracted four times with 40 mL of diethyl ether (Et_2_O). The combined ether extracts were dried with MgSO_4_ and the solvent was evaporated. The resultant solid crude product was dryed under vacuum and then purified by silica gel (230–400 mesh) column chromatography using CH_3_CN:CHCl_3_ 1:4 as eluting solvent system. **Yield**: 320 mg (91%); **TLC** (CH_3_CN:CHCl_3_ 1:2); **R_f_**: 0.35; **^1^H NMR** (500 MHz, CD_3_CN) δ: 7.47, 6.07 (1H, *P-H*), 3.70 (s, 4H, *CH_carborane_*), 3.10 (m, 6H, N*CH_2_*CH_3_), and 1.24 (t, 9H, NCH_2_*CH_3_*); **^13^C{^1^H} NMR** (125 MHz, CD_3_CN) δ: 47.96 (4C, C_carb_), 47.74 (3C, N*CH_2_*CH_3_), and 9.24 (3C, NCH_2_*CH_3_*); **^11^B{^1^H} NMR** (120 MHz, CD_3_CN) δ: 23.02 (s, 2B, B^8,8′^), −2.83 (s, 2B, B^10,10′^), −5.70 (s, 4B, B^9,9′,12,12′^), −7.94 (s, 2B, B^4,4′^), −8.74 (s, 2B, B^7,7′^), −18.89 (s, 4B, B^5,5′,11,11′^), and −27.85 (s, 2B, B^6,6′^); **^11^B NMR** (125 MHz, CD_3_CN) δ: 23.02 (s, 2B, B^8,8′^), −2.83 (d, 2B, B^10,10′^), −5.70 (d, 4B, B^9,9′,12,12′^), −8.36 (t, 4B, B^4,4′,7,7′^), −18.91 (d, 4B, B^5,5′,11,11′^), and −27.84 (d, 2B, B^6,6′^); **^31^P{^1^H} NMR** (202 MHz, CD_3_CN) δ: −3.01 (s, P-H); **^31^P NMR** (202 MHz, CD_3_CN) δ: −3.00 (d, P-H); **ATR-IR (cm^−1^):** 3621, 3029, 2993, 2544, 1609, 1474, 1446, 1393, 1218, 1152, 1137, 1094, 1025, 993, 981, 920, 903, 871, 849, 787, 743, 691, and 666. **UV-Vis** λ_max_ (nm): 297 and 445. **ESI-MS** (*m*/*z*)**:** found 402.24 (calc. for C_4_H_21_O_3_B_18_P_1_Co_1_: 401.71).

*Synthesis of 8,8′-bridged [8,8′-O_2_P(O)SH-3,3′-Co(1,2-C_2_B_9_H_10_)_2_] HDBU phosphorothioate (***13***).* H-phosphonate acid ester **12** (130 mg, 0.26 mmol) was dissolved in anhydrous MeOH (6.5 mL). The solution was added under argon atmosphere to S_8_ (85 mg, 2.6 mmol). Then, 1,8-Diazabicyclo(5.4.0)undec-7-en (DBU) (160 µL, 1.05 mmol) was added, and the mixture was stirred for 96 h at room temperature. Subsequently, solvent was evaporated under reduced pressure. The crude product was dissolved in CH_3_CN and then purified via silica gel column chromatography using a gradient of CH_3_CN:CHCl_3_ from 1:4 to 1:1 as eluting solvent system. Finally, product **13** was eluted from the column using 100% MeOH as eluent. **Yield:** 105 mg (70%); **TLC** (CH_3_CN:CHCl_3_ 2:1) **R_f_:** 0.5; **^1^H NMR** (500 MHz, CD_3_CN) δ: 9.14 (s, 1H, *NH*), 3.58 (s, 4H, *CH_carborane_*), 3.50 (m, 2H, NH*CH_2_*CH_2_), 3.44 (t, *J* = 5.9 Hz, 2H, CH_2_N*CH_2_*CH_2_CH_2_NH), 3.31 (s, 2H, *CH_2_*NCH_2_CH_2_CH_2_NH), 2.69 (dd, *J* = 6.6, 3.5 Hz, 2H, NHC*CH_2_*), 1.97 (dd, *J* = 6.6 Hz, 2H, CH_2_NCH_2_*CH_2_*CH_2_NH), 1.72 (m, 4H, NHCCH_2_*CH_2_CH_2_*), and 1.65 (dt, *J* = 14.9, 5.2 Hz, 2H, NHCCH_2_CH_2_CH_2_*CH_2_*); **^13^C{^1^H} NMR** (125 MHz, CD_3_CN) δ: 166.94 (NH*C*CH_2_), 54.98 (*CH_2_*NCH_2_CH_2_CH_2_NH), 49.26 (CH_2_N*CH_2_*CH_2_CH_2_NH), 46.76 (C_carb_), 46.63 (C_carb_), 39.04 (NH*CH_2_*), 33.42 (NHC*CH_2_*), 29.47 (NHCCH_2_*CH_2_*CH_2_), 27.05 (NHCCH_2_CH_2_CH_2_*CH_2_*), 24.52 (NHCCH_2_CH_2_*CH_2_*), and 19.96 (CH_2_NCH_2_*CH_2_*CH_2_NH); **^11^B{^1^H} NMR** (120 MHz, CD_3_CN) δ: 23.46 (s, 2B, B^8,8′^), −3.54 (s, 2B, B^10,10′^), −5.72 (s, 4B, B^9,9′,12,12′^), −8.73 (s, 4B, B^4,4′,7,7′^), −19.49 (s, 4B, B^5,5′,11,11′^), and −28.25 (s, 2B, B^6,6′^); **^11^B NMR** (120 MHz, CD_3_CN) δ: 23.46 (s, 2B, B^8,8′^), −3.50 (d, 2B, B^10,10′^), −5.74 (d, 4B, B^9,9′,12,12′^), −8.71 (d, 4B, B^4,4′,7,7′^), −19.53 (d, 4B, B^5,5′,11,11′^), and −28.34 (d, 2B, B^6,6′^); **^31^P{^1^H}NMR** (202 MHz, CD_3_CN) δ: 48.63 (s); **^31^P NMR** (202 MHz, CD_3_CN) δ: 48.62 (s); **ATR-IR (cm^−1^):** 3383 (ν OH), 3223 (ν NH), 3091 (ν NH), 3026, 2926 (ν CH), 2856 (ν CH), 2799 (ν CH), 2545 (ν BH), 1725, 1640, 1607, 1465, 1444, 1363, 1321, 1292, 1205, 1157, 1103, 1076, 978, 936, 910, 887, 836, 747, and 690. UV-Vis λ_max_ (nm) 215, 296, and 450. **ESI-MS** (*m*/*z*): found: 434.20 (calc. for C_4_H_21_O_3_B_18_P_1_S_1_Co_1_ 433.78).

*Synthesis of 8,8′-bridged [8,8′-O_2_P(O)S(CH_2_)_n_OCPh_3_-3,3′-Co(1,2-C_2_B_9_H_10_)_2_] HNEt_3_ S-alkylated phosphorothioates* **15** *and* **16**. [8,8′-O_2_P(O)SH-3,3′-Co(1,2-C_2_B_9_H_10_)_2_] HDBU (**13**) (13 mg, 0.02 mmol) was dissolved in acetone (0.520 mL); then, Et_3_N (65 µL, 0.46 mmol) was added under stirring at room temperature. The mixture was heated to 60 °C in an oil bath; then, alkylating agent **3** or **14** (0.04 mmol, dissolved in 130 µL of CH_2_Cl_2_) was added. The reaction mixture was maintained overnight at 60 °C with stirring; then, it was cooled to room temperature, and solvents were evaporated under reduced pressure. The residue was dispersed in CH_2_Cl_2_ and then filtered and the solution was loaded on silica gel column prepared in CH_2_Cl_2_. Chromatography was performed using a gradient of MeOH in CH_2_Cl_2_ from 0 to 3% MeOH. **(15): Yield:** 17 mg (90%); **TLC** (CH_3_CN:CHCl_3_ 1:2): **R_f_:** 0.71; **^1^H NMR** (500 MHz, CD_3_CN): δ 7.41 (d, *J* = 7.5 Hz, 6H, *H_arom_*), 7.31 (t, *J* = 7.6 Hz, 6H, *H_arom_*), 7.23 (t, *J* = 7.2 Hz, 3H, *H_arom_*), 3.67 (s, 4H, *CH_carborane_*), 3.07 (q, *J* = 7.3 Hz, 6H, N*CH_2_*CH_3_), 3.00 (t, *J* = 6.0 Hz, 2H SCH_2_CH_2_CH_2_*CH_2_*OTr), 2.78 (dt, 2H, S*CH_2_*CH_2_CH_2_CH_2_OTr), 1.72 (m, *J* = 7.1 Hz, 2H SCH_2_CH_2_*CH_2_*CH_2_OTr), 1.64 (m, *J* = *6.9* Hz, 2H, SCH_2_*CH_2_*CH_2_CH_2_Otr), and 1.21 (t, *J* = 7.3 Hz, 9H, NCH_2_*CH_3_*); **^13^C{^1^H} NMR** (126 MHz, CD_3_CN): δ 145.55 (3C, *aromatic_trityl_*), 129.54 (6C, *aromatic_trityl_*), 128.87 (6C, *aromatic_trityl_*), 128.01 (3C, *aromatic_trityl_*), 87.30 (1C, O*C(Ph)_3_*), 63.90 (1C, SCH_2_CH_2_CH_2_*CH_2_*OTr), 47.76 (overlapped 3C, HN*CH_2_*CH_3_, 4C, *CH_carborane_*), 31.20 (1C, S*CH_2_*CH_2_CH_2_CH_2_OTr), 29.75 (1C, SCH_2_CH_2_*CH_2_*CH_2_OTr), 28.84 (1C, SCH_2_*CH_2_*CH_2_CH_2_OTr), and 9.29 (1C, NCH_2_*CH_3_*); **^11^B{^1^H} NMR** (160 MHz, CD_3_CN): δ 23.08 (s, 2B, B^8,8′^), −2.86 (s, 2B, B^10,10′^), −5.54 (s, 4B, B^9,9′,12,12′^), −8.36 (s, 4B, B^4,4′,7,7′^), −18.87 (s, 4B, B^5,5′,11,11′^), and −27.70 (s, 2B, B^6,6′^); **^11^B NMR** (160 MHz, CD_3_CN): δ 23.06 (s, 2B, B^8,8′^), −2.87 (d, 2B, B^10,10′^), −5.57 (d, 4B, B^9,9′,12,12′^), −8.16 (d, 4B, B^4,4′,7,7′^), −18.92 (d, 4B, B^5,5′,11,11′^), and −27.63 (d, 2B, B^6,6′^); **^31^P NMR{^1^H}** (202 MHz, CD_3_CN): δ 16.72 (s), 11.37 (s); **^31^P NMR** (202 MHz, CD_3_CN): δ 16.72 (t) and 11.37 (t); **ATR-IR (cm^−1^)** 3032 (ν CH aromatic), 2987 (ν CH aliphatic), 2925 (ν CH aliphatic), 2851 (ν CH aliphatic), 2681, 2566 (ν BH), 1727, 1595, 1474, 1447, 1392, 1264, 1200, 1134, 1103, 1068, 1032, 1016, 980, 941, 917, 901, 849, 735 (aromatic CH bending), and 705 (aromatic CH bending). **UV-Vis** λ_max_ (nm) 196, 299, and 440. **ESI-MS** (*m*/*z*): found: 748.37 (calc. for C_27_B_18_H_43_O_4_P_1_S_1_Co_1_: 748.24).

(**16**): **Yield:** 16 mg (86%); **TLC** (CH_3_CN:CHCl_3_ 1:2) **Rf**: 0.69; **^1^H NMR** (500 MHz, CD_3_CN) δ: 7.43 (m, 6H, *H_arom_*), 7.33 (m, 6H, *H_arom_*), 7.25 (m, 3H, *H_arom_*), 3.67 (s, 4H, *CH_carborane_*), 3.10 (t, *J* = 6.0 Hz, 2H SCH_2_CH_2_*CH_2_*OTr), 3.08 (q, *J* = 7.3 Hz, 6H, N*CH_2_*CH_3_), 2.97 (dt*, J* = *14.9*, 7.3 Hz, 2H, PS*CH_2_*CH_2_CH_2_OTr), 2.58 (s, 1H, *CH_3_OH*), 1.27 (t, *J* = 5.4 Hz, 2H PSCH_2_*CH_2_*CH_2_OTr), 1.22 (t, *J* = 7.3 Hz, 9H, NCH_2_*CH_3_*), and 1.18 (s, 3H, CH_3_OH); **^13^C{^1^H} NMR** (126 MHz, CD_3_CN) δ: 145.27 (3C, *aromatic_trityl_*), 129.45 (6C, *aromatic_trityl_*), 128.78 (6C, *aromatic_trityl_*), 127.94 (3C, *aromatic_trityl_*), 87.25 (O*C(Ph)_3_*), 62.71 (PSCH_2_CH_2_*CH_2_*OTr), 55.13 (4C, *CH_carb_*), 47.64 (HN*CH_2_*CH_3_), 47.59, 32.14, 31.98 (d, *J* = 6.1 Hz), 30.29, 29.66, 28.41 (d, *J* = 3.8 Hz), and 9.17 (HNCH_2_*CH_3_*); **^11^B{^1^H} NMR** (160 MHz, CD_3_CN): δ (ppm) 23.08 (s, 2B, B^8,8′^), −2.86 (s, 2B, B^10,10′^), −5.53 (s, 4B, B^9,9′,12,12′^), −8.36 (s, 4B, B^4,4′,7,7′^), −18.87 (s, 4B, B^5,5′,11,11′^), and −27.70 (s, 2B, B^6,6′^); **^11^B NMR** (160 MHz, CD_3_CN): δ 23.07 (s, 2B, B^8,8′^), −2.92 (d, 2B, B^10,10′^), −5.60 (d, 4B, B^9,9′,12,12′^), −8.38 (d, 4B, B^4,4′,7,7′^), −18.98 (d, 4B, B^5,5′,7,7′^), and −27.88 (d, 2B, B^6,6′^); **^31^P{^1^H} NMR** (202 MHz, CD_3_CN) δ: 16.54 (s); **^31^P NMR** (202 MHz, CD_3_CN) δ: 16.54 (t); **ATR-IR (cm^−1^)** 3032 (ν CH aromatic), 2987 (ν CH aliphatic), 2925 (ν CH aliphatic), 2851 (ν CH aliphatic), 2684, 2567 (ν BH), 1699, 1596, 1474, 1447, 1392, 1265, 1200, 1135, 1104, 1066, 1032, 1016, 980, 941, 917, 902, 849, 735 (aromatic CH bending), and 705 (aromatic CH bending). **UV-Vis** λ_max_ (nm) 194, 299, and 436. **ESI-MS** (*m*/*z*): 734.35 (calc. for C_26_B_18_H_41_O_4_P_1_S_1_Co_1_ 734.17).

*Synthesis of 4-[1,3-bis(trityloxy)propan-2-yl-oxy]butyl-4-methylbenzenesulfonate (***20***).* The reaction was performed under argon atmosphere in anhydrous conditions. 1,3-Bis(trityloxy)propan-2-ol (**18**) (2.35 g, 4.07 mmol) was dissolved in 18 mL of anhydrous DMF; then, NaH_60%_ (195 mg, 4.87 mmol) was added. After stirring for 15 min, 1,4-bis(*p*-toluenesulfonyloxy)butane [40] (4.23 g, 10.63 mmol), dissolved in 18 mL of DMF, was added. The reaction mixture was stirred for another 2 h at room temperature and was then cooled in ice bath; subsequently, an excess of NaH was centrifuged. The supernatant was poured into a cooled 40 mL volume of phosphate buffer. The mixture was extracted with AcOEt (4x 100 mL). Organic extracts were combined, washed with H_2_O, and dried over MgSO_4._ Solvents were evaporated under reduced pressure. The crude product was purified via silica gel column chromatography using a gradient of AcOEt in hexane from 0% to 10% as eluting solvent system. **Yield:** 883 mg (27%); **TLC** (Hexane:AcOEt 2:1) **R_f_:** 0.55; **^1^H NMR** (600 MHz, CDCl_3_) δ: 7.78 (d, 2H, *H_arom_*), 7.41 (d, 12H, *H_arom_*), 7.27 (m, 20H, *H_arom_*), 4.04 (t, 2H, OCH_2_CH_2_CH_2_*CH_2_*OSO_2_), 3.55 (p, 1H, TrOCH_2_*CHO*CH_2_OTr), 3.48 (t, 2H, O*CH_2_*CH_2_CH_2_CH_2_OSO_2_), 3.23 (ddd, 4H, TrO*CH_2_*CH), 2.43 (s, 3H, *CH_3tosyl_*), 1.75 (m, 2H, OCH_2_CH_2_*CH_2_*CH_2_OSO_2_), and 1.58 (m, 2H, OCH_2_*CH_2_*CH_2_CH_2_OSO_2_); **^13^C{^1^H} NMR** (125 MHz, CD_3_CN) δ: 144.62 (1C, *aromatic_tosyl_*), 144.05 (6C, *aromatic_trityl_*), 133.19 (1C, *aromatic_tosyl_*), 129.81 (2C, *aromatic_tosyl_*), 128.73 (12C, *aromatic_trityl_),* 127.88 (2C, *aromatic_tosyl_*), 127.76 (12C, *aromatic_trityl_*), 126.91 (6C, *aromatic_trityl_*), 86.52 (2C, O*C(Ph)_3_*), 78.59 (1C, TrOCH_2_*CHO*CH_2_OTr), 70.51 (1C, OCH_2_CH_2_CH_2_CH_2_OSO_2_-), 69.47 (1C, O*CH_2_*CH_2_CH_2_CH_2_OSO_2_-), 63.39 (2C, TrO*CH*_2_CH), 26.10 (1C, OCH_2_CH_2_*CH_2_*CH_2_OSO_2_), 25.86 (1C, OCH_2_*CH_2_*CH_2_CH_2_OSO_2_), and 21.64 (1C, CH_3_^tosyl^); **ATR-IR (cm^−1^):** 3054, 3018, 2946, 2929, 2869, 1978, 1732, 1596, 1488, 1447, 1352, 1304, 1218, 1187, 1172, 1122, 1094, 1065, 1029, 992, 950, 922, 841, 811, 768, 745, and 699; **UV-Vis** λ_max_ (nm): 198, 229, and 260; **ESI**-**MS** (*m*/*z***):** found 825.32 [M + Na]^−^, 841.29 [M + K]^−^, (calc. for C_52_H_50_O_6_S_1_ 803.01).

*Synthesis of 8,8′-bridged {8,8′-O_2_P(O)S[(CH_2_)_4_OCH(CH_2_OCPh_3_)_2_]-3,3′-Co(1,2-C_2_B_9_H_10_)_2_} HNEt_3_ (***21***).* [8,8′-O_2_P(O)SH-3,3′-Co(1,2-C_2_B_9_H_10_)_2_] HDBU (**13**) (440 mg, 0.75 mmol) was dissolved in 20 mL of anhydrous acetone; then, anhydrous Et_3_N (2.15 mL, 15.42 mmol) was added to the resultant solution under stirring at room temperature. The mixture was heated to 60 °C; then, 4-[1,3-bis(trityloxy)propan-2-yl-oxy]butyl-4-methylbenzenesulfonate (**20**) (901 mg, 1.12 mmol), which was dissolved in 20 mL of anhydrous AcOEt, was added dropwise. After stirring overnight at 60 °C, the mixture was cooled, and the solvents were evaporated under reduced pressure. The residue was dispersed in CH_2_Cl_2_, filtered, and the filtrate was loaded into the silica gel column prepared in CH_2_Cl_2_. Chromatography was performed using a gradient of CH_3_OH in CH_2_Cl_2_ from 0 to 3% of CH_3_OH. **Yield:** 560 mg (64%); **TLC** (CH_3_CN:CHCl_3_ 1:2), **R_f_**: 0.60; **^1^H NMR** (600 MHz, CD_3_CN) δ: 7.41 (d, 12H, *H_aromatic_*), 7.31 (m, 18H, *H_aromatic_*), 3.70 (s, 4H, *CH_carborane_*), 3.60 (p, 1H, TrOCH_2_*CHO*CH_2_OTr), 3.45 (t, 2H, PSCH_2_CH_2_CH_2_*CH_2_*O), 3.19 (ddd, 4H, CHO*CH_2_*OTr), 3,10 (q, 6H, HN*CH_2_*CH_3_), 2.85 (dt, 2H, PS*CH_2_*CH_2_CH_2_CH_2_O), 1.71 (m, 2H, PSCH_2_CH_2_*CH_2_*CH_2_O), 1.61 (m, 2H, PSCH_2_*CH_2_*CH_2_CH_2_O), and 1.24 (t, 9H, NCH_2_*CH_3_*); **^13^C{^1^H} NMR** (125 MHz, CD_3_CN) δ: 144.73 (6C, *aromatic_trityl_*), 129.10 (12C, *aromatic_trityl_*), 128.41 (12C, *aromatic_trityl_*), 127.60 (6C, *aromatic_trityl_*), 86.89 (2C, O*C(Ph)_3_*), 78.47 (1C, TrOCH_2_*CHO*CH_2_OTr), 69.92 (1C, PSCH_2_CH_2_CH_2_*CH_2_*O), 63.66 (2C, CHO*CH_2_*OTr), 47.28 (3C, HN*CH_2_*CH_3_), 47.18 (4C, *CH_carborane_*), 31.89 (1C, PS*CH_2_*CH_2_CH_2_CH_2_O), 22.93 (1C, PSCH_2_CH_2_*CH_2_*CH_2_O), 13.96 (1C, PSCH_2_*CH_2_*CH_2_CH_2_O), and 8.80 (3C, HNCH_2_*CH_3_)*; **^11^B{^1^H} NMR** (120 MHz, CD_3_CN) δ: 23.02 (s, 2B, B^8,8′^), −3.04 (s, 2B, B^10,10′^), −5.67 (s, 4B, B^9,9′,12,12′^), −8.55 (s, 4B, B^4,4′,7,7′^), −19.07 (s, 4B, B^5,5′,11,11′^), and −27.99 (s, 2B, B^6,6′^); **^11^B NMR** (120 MHz, CD_3_CN) δ: 23.02 (s, 2B, B^8,8′^), −2.98 (d, 2B, B^10,10′^), −5.64 (d, 4B, B^9,9′,12,12′^), −8.19 (d, 4B, B^4,4′,7,7′^), −19.05 (d, 4B, B^5,5′,11,11′^), and −27.99 (d, 2B, B^6,6′^); ^31^P{^1^H}NMR (202 MHz, CD_3_CN) δ (ppm) 16.49 (s); **^31^P NMR** (202 MHz, CD_3_CN) δ: 16.48 (t); **ATR-IR (cm^−1^):** 3031, 2929, 2870, 2564, 2564, 2161, 1978, 1644, 1595, 1489, 1447, 1322, 1202, 1134, 1096, 1031, 940, 899, 848, 763, 747, and 697; **UV-Vis** λ_max_ (nm): 196, 233, 299, and 453; **MS (ESI)** (*m*/*z*): found 1064.52 (calc. for C_49_B_18_H_63_O_6_P_1_S_1_Co_1_ 1064.59).

*Synthesis of 8,8′-bridged {8,8′-O_2_P(O)S[(CH_2_)_4_OCH(CH_2_OCPh_3_)_2_]-3,3′-Co[1-(CH_2_)_2_OH-1,2-C_2_B_9_H_10_)](1′,2′-C_2_B_9_H_10_)} HNEt_3_ (**22**) and {8,8′-O_2_P(O)S[(CH_2_)_4_OCH(CH_2_OCPh_3_)_2_]-3,3′-Co[1-(CH_2_)_2_OH-1,2-C_2_B_9_H_10_)] [1′-(CH_2_)_2_OH-1′,2′-C_2_B_9_H_10_)]} HNEt_3_ (***23***).* Compound **21** (175 mg, 0.15 mmol) was dried via co-evaporation with anhydrous benzene and then kept under vacuum over P_2_O_5_ overnight. Then, it was dissolved in anhydrous DME (3 mL), and the solution was cooled in CO_2_/isopropanol cooling bath. After 15 min, n-BuLi (140 µL, 1.6 M solution in hexane, 1.5 eq) was added, and the reaction mixture was stirred for 10 min. Afterwards, the cooling bath was removed, and the mixture was stirred for next 10 min. Then, the reaction mixture was cooled again in cooling bath and another portion of n-BuLi (140 µL) was added. After 15 min, ethylene oxide (200 µL, 2.9–3.1 M solution in THF)) was added, and the reaction was left overnight in cooling bath. Then, CH_2_Cl_2_ (5 mL) was added to the reaction mixture, the reaction was quenched via addition of water, and then the organic solution was washed three times with 5 mL portions of water. Organic layer was separated and dried over MgSO_4_; then, solvents were evaporated. Crude product was purified, and mono- and bis-substituted products were separated via silica gel column chromatography using a gradient of MeOH in CH_2_Cl_2_ from 0 to 3% of MeOH. (**22)**: **Yield**: 17 mg (10%); **TLC** (MeOH:CH_2_Cl_2_ 1:12.5): **R_f_**: 0.13; **^1^H NMR** (500 MHz, CD_3_CN): δ (ppm) 7.41 (d, 12H, *H_aromatic_*), 7.31 (m, 18H, *H_aromatic_*), 3.90 (s, *CH_carborane_*), 3.78 to 3.54 (m, overlapped, *CH_carborane_*, HO*CH_2_*CH_2_C_carb_, TrOCH_2_*CHO*CH_2_OTr), 3.44 (t, 2H, PSCH_2_CH_2_CH_2_*CH_2_*O-), 3.19 (ddd, 4H, CHO*CH*_2_OTr), 3.02 to 2.64 (m, overlapped, PS*CH_2_*CH_2_CH_2_CH_2_O-, HOCH_2_*CH_2_C_carb_*), and 1.65 (m, 4H, overlapped, PSCH_2_*CH_2_CH_2_*CH_2_O); ^**11**^**B{^1^H} NMR** (120 MHz, CD_3_CN) δ (ppm) 25.37, 24.65, 23.69, 22.93 (in ratio 2:2:1:1) 26.6 to 21.37 (m, overlapped diastereoizomeric B^8,8′^), 0.52 to −11.61 (m, overlapped diastereoizomeric, B^10,10′,9,9′,12,12′,4,4′,7,7′^), −12.05 to −11.61 (d, overlapped diastereoizomeric B^5,5′,11,11′^), and −21.45 to −28.25 (s, overlapped diastereoizomeric B^6,6′^); **^11^B NMR** (120 MHz, CD_3_CN) δ (ppm) 27.17 to 21.57 (m, overlapped diastereoizomeric B^8,8′^), 2.85 to −11.77 (m, overlapped diastereoizomeric, B^10,10′,9,9′,12,12′,4,4′,7,7′^), −11.82 to −21.00 (d, overlapped diastereoizomeric B^5,5′,11,11′^), and −21.10 to −29.92 (s, overlapped diastereoizomeric B^6,6′^); **^31^P{^1^H}NMR** (202 MHz, CD_3_CN) δ: 14.94, 14.58, 14.43, 14.12, and 13.49 (in ratio: 4:1:2:1.5:15); **^31^P NMR** (202 MHz, CD_3_CN) δ: 14.94 (t), 14.42 (t), 14.12 (t), 13.49 (t); **ATR-IR (cm^−1^):** 3630, 3370, 3057, 3031, 2925, 2869, 2565, 2161, 1979, 1596, 1489, 1448, 1255, 1202, 1128, 1077, 1032, 985, 898, 871, 763, 746, and 699; **ESI-MS** (*m*/*z*): found: 1108.54 *m*/*z* (calc. for C_51_B_18_H_67_O_7_P_1_S_1_Co_1_ 1108.64).

**(23)**: **Yield:** 21 mg (12%); **TLC** (MeOH:CH_2_Cl_2_ 1:12.5): **R_f_:** 0.27; **^1^H NMR** (500 MHz, CD_3_CN) δ: 7.40 (d, 12H, *H_aromatic_*), 7.31 (m, 18H, *H_aromatic_*), 3.84 to 3.51 (m, overlapped, *CH_carborane_*, HO*CH_2_*CH_2_-, TrOCH_2_*CHO*CH_2_OTr,), 3.44 (t, 2H, PSCH_2_CH_2_CH_2_*CH_2_*O-), 3.18 (ddd, 4H, CHOCH_2_OTr), 3.12 to 2.66 (m, overlapped, PS*CH_2_*CH_2_CH_2_CH_2_O, HOCH_2_*CH_2_*C_carborane_*,*), and 1.65 to 1.55 (m, 4H, overllaped, PSCH_2_*CH_2_CH_2_*CH_2_O); ^**11**^**B{^1^H} NMR** (120 MHz, CD_3_CN) δ: 25.36, 24.41, 23.65, 22.80, 22.23 (in ratio 1:2:1.5:1:1), 26.59 to 20.45 (m, overlapped diastereoizomeric B^8,8′^), 2.75 to −12.40 (m, overlapped diastereoizomeric, B^10,10′,9,9′,12,12′,4,4′,7,7′^), and −12.34 to −24.64 (m overlapped diastereoizomeric B^5,5′,11,11′6,6′^); **^11^B NMR** (120 MHz, CD_3_CN) δ: 26.65 to 20.21 (m, overlapped diastereoizomeric B^8,8′^), 2.32 to −12.88 (m, overlapped diastereoizomeric, B^10,10′,9,9′,12,12′,4,4′,7,7′^), −12.84 to −25.84 (m overlapped diastereoizomeric B^5,5′,11,11′6,6′^); **^31^P{^1^H}NMR** (202 MHz, CD_3_CN) δ: 15.00, 14.16, 14.02, 13.38 (in ratio 5:1:3:1); **^31^P NMR** (202 MHz, CD_3_CN) δ: 15.99 (t), 14.40 (s), 14.01 (t), and 13.38 (t); **ATR-IR (cm^−1^):** 3566, 3357, 3056, 3027, 2920, 2889, 2857, 2565, 2166, 1596, 1489, 1448, 1291, 1255, 1201, 1120, 1078, 1032, 1001, 889, 871, 764, 746, and 699 **ESI-MS** (*m*/*z*): found: 1152.57 (calc. for C_53_B_18_H_71_O_8_P_1_S_1_Co 1152.69).

## 5. Conclusions

Although derivatives of metallocarboranes containing various simple substituents attached to the boron or carbon atoms of the complex carboranyl ligands are abundant, they do not usually allow for further chemical transformations. The adducts of some metallacarboranes and cyclic ethers are notable exceptions. In this work, we proposed methods for filling this gap, at least partially, by using extendable ligands.

The exploitation of icosahedral metallacarborane’s immense potential in various fields of chemistry and technology requires the availability of convenient and versatile methods for their modification with various functional moieties and/or linkers of various types and lengths. Herein, we report a convenient approach to introducing extendible arms on 8,8′-dihydroxy cobalt bis(1,2-dicarbollide). This approach can be used to introduce different hetero-bifunctional electrophiles containing a protected hydroxyl function that allows for further modification.

## Data Availability

Data is contained within the article or Appendix A.

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
