# Peer review of "Metallacarborane Synthons for Molecular Construction—Oligofunctionalization of Cobalt Bis(1,2-dicarbollide) on Boron and Carbon Atoms with Extendable Ligands"

_molecules, 2023, doi:10.3390/molecules28104118_

Round 1
Reviewer 1 Report
1. The title cannot highlight your full content.
2. How about the yield for each compound?
3. I suggest the authors could list a Table for the full work.
4. Check the typesetting work.
revision
Author Response
- The title cannot highlight your full content.
To highlight the content of the manuscript more comprehensively the title was changed from “Metallacarborane Synthons for Molecular Construction - Functionalization of Cobalt bis(1,2-dicarbollide) with Extendable Ligands” to “Metallacarborane Synthons for Molecular Construction - Oligofunctionalization of Cobalt bis(1,2-dicarbollide) on Boron and Carbon atoms with Extendable Ligands”
- How about the yield for each compound?
The yield (marked in bold) for each new compound are provided in Materials and Methods section.
- I suggest the authors could list a Table for the full work.
The yield of synthesis for each compound is conveniently shown at the end of each synthesis procedure, along with its characteristics in Materials and Methods section, according to the generally used layout. In order to keep the text concise, we decided not to repeat this data in the table.
- Check the typesetting work.
Text of the manuscript was checked with typesetting, grammar, and spelling correction software.
Reviewer 2 Report
In this work, Śmiałkowski et al presented the synthesis and spectroscopic characterizations of a series of Cobalt-based metallacarborane clusters. It is a solid and well-prepared work that is worth publication in Molecules due to the unique structural, properties and potential applications of boron clusters.
This manuscript is well prepared and I only listed a few format issues for the authors’ further consideration, mostly related to font types, spaces, etc.
Lines 32, 88, 104, 144, 319, and 337.
Data presentation format inconsistency: Lines 358/361 versus 398/417 versus 475/478 versus 678.
References: Inconsistent capitalization of paper titles; incorrect presentation of journal names for those journals without abbreviation, such as Nanoscale and Molecules; inconsistent format for the pages, such as Ref. 40 versus others.
Author Response
- This manuscript is well prepared and I only listed a few format issues for the authors’ further consideration, mostly related to font types, spaces, etc. Lines 32, 88, 104, 144, 319, and 337.
The size and font types are automatically imposed by the template, the spaces were checked and corrected.
- Data presentation format inconsistency: Lines 358/361 versus 398/417 versus 475/478 versus 678.
The data presented in the 4.2. Methods section has been unified, protons/carbons corresponding to a specific chemical shift are now consistently underlined. TLC data, yields description, was also unified.
- References: Inconsistent capitalization of paper titles; incorrect presentation of journal names for those journals without abbreviation, such as Nanoscale and Molecules; inconsistent format for the pages, such as Ref. 40 versus others.
The references’ format was unified.
Reviewer 3 Report
Minor revision The paper is interesting, well-ogranized and of great scientific importance for chemists dealing with carbaborane chemistry. There are a lot of ways for chemical design and methods used by the authors can be used for other carborane-containing objects. It can be published after some corrections and remarks. Please, add relevant references to the INTRODUCTION https://doi.org/10.1134/S0036023622600848 https://doi.org/10.1016/j.ccr.2022.214636 https://doi.org/10.1016/j.ccr.2019.213139 https://doi.org/10.1016/j.ccr.2020.213684 https://doi.org/10.1134/S0036023621090151 The compounds under the study have a great potential to be used in varous fields. Did the authors perform any experiments that can demonstrate the possibility of the application of their new compounds in any field? It’s a pity that none of compounds was identified with X-ray diffraction data. Are the compounds difficult to be crystallized? Can the single-crystal be received? Are you sure about the composition and structure of compound 5? Can compound 5 have another structure? Minor revision The paper is interesting, well-ogranized and of great scientific importance for chemists dealing with carbaborane chemistry. There are a lot of ways for chemical design and methods used by the authors can be used for other carborane-containing objects. It can be published after some corrections and remarks. Please, add relevant references to the INTRODUCTION https://doi.org/10.1134/S0036023622600848 https://doi.org/10.1016/j.ccr.2022.214636 https://doi.org/10.1016/j.ccr.2019.213139 https://doi.org/10.1016/j.ccr.2020.213684 https://doi.org/10.1134/S0036023621090151 The compounds under the study have a great potential to be used in varous fields. Did the authors perform any experiments that can demonstrate the possibility of the application of their new compounds in any field? It’s a pity that none of compounds was identified with X-ray diffraction data. Are the compounds difficult to be crystallized? Can the single-crystal be received? Are you sure about the composition and structure of compound 5? Can compound 5 have another structure?Author Response
INTRODUCTION https://doi.org/10.1134/S0036023622600848 https://doi.org/10.1016/j.ccr.2022.214636 https://doi.org/10.1016/j.ccr.2019.213139 https://doi.org/10.1016/j.ccr.2020.213684 https://doi.org/10.1134/S0036023621090151
Thank you for sending further examples illustrating the wide range of applications of boron clusters. Due to the focus of this paper on metallacarboranes type of cobalt bis(1,2-dicarbollide), we have decided to add the paper Ann. N.Y. Acad. Sci., 2019, 1457, 128–141 to the list of references, in the section on applications in medicinal chemistry, as a reference 4b.
- The compounds under the study have a great potential to be used in varous fields. Did the authors perform any experiments that can demonstrate the possibility of the application of their new compounds in any field?
We thank the reviewer for this question. Oligofunctionalized boron clusters, including their complexes with metals, metallacarboranes, can and do find several applications. The subject of our interest is the use of composites of boron clusters and DNA as building blocks for the construction of nanoparticles and their use as carriers of therapeutic nucleic acids. An example of this is our publication in Nanoscale (2020, 12, 103–114). Another paper presenting the second generation of nanoparticles of this type containing carborane as a scaffold is in preparation for publication. Work is advanced on the construction and use of similar nanoparticles based on the structures of oligofunctionalized metallacarboranes, such as described in this work, with DNA attached. Research on the automatic, chemical synthesis of DNA on an oligofunctionalized cobalt-containing metallacarborane has been successful and will be published soon.
- It’s a pity that none of the compounds was identified with X-ray diffraction data. Are the compounds difficult to be crystallized? Can the single-crystal be received?
X-ray diffraction is a method commonly used in the study of boron clusters, especially their simpler derivatives. In the case of more complex compounds, such as conjugates of boron clusters and biomolecules or oligofunctionalized boron cluster derivatives, obtaining crystals suitable for X-ray analysis is a major challenge. For example, although the chemistry of nucleosides and boron clusters conjugates is well developed, only three X-ray structures of this type of molecules have been published so far (http://dx.doi.org/10.1039/C4NJ01096E; doi.org/10.1016/j.molstruc.2022.134588).
- Are you sure about the composition and structure of compound 5? Can compound 5 have another structure?
The composition of compound 5 was proved by MS and NMR analyses. Concerning the structure, metallacarboranes can rotate around the vertical axis, in Scheme 1, only the conformer trans is shown. There are possible other conformers, including the opposite one, conformer cis. Because without detailed studies it is impossible to determine which conformer we are dealing with (most likely it is a mixture of conformers with one in predominance) for clarity, the trans conformer which possibly is thermodynamically preferred, is shown. Albeit, a cis conformer stabilized by a hydrogen bond between the hydrogen atom of the hydroxyl group and the oxygen atom of the alkylated hydroxyl group cannot be excluded.